# Online Bidding under RoS Constraints
# without Knowing the Value

## Abstract

We consider the problem of bidding in online advertising, where an advertiser aims to maximize value while adhering to budget and Return-on-Spend (RoS) constraints. Unlike prior work that assumes knowledge of the value generated by winning each impression (e.g., conversions), we address the more realistic setting where the advertiser must simultaneously learn the optimal bidding strategy and the value of each impression opportunity. This introduces a challenging exploration-exploitation dilemma: the advertiser must balance exploring different bids to estimate impression values with exploiting current knowledge to bid effectively. To address this, we propose a novel Upper Confidence Bound (UCB)-style algorithm that carefully manages this trade-off. Via a rigorous theoretical analysis, we prove that our algorithm achieves $\widetilde{O}(\sqrt{T \log(|\mathcal{B}|T)})$ regret and constraint violation, where $T$ is the number of bidding rounds and $\mathcal{B}$ is the domain of possible bids. This establishes the first optimal regret and constraint violation bounds for bidding in the online setting with unknown impression values. Moreover, our algorithm is computationally efficient and simple to implement. We validate our theoretical findings through experiments on synthetic data, demonstrating that our algorithm exhibits strong empirical performance compared to existing approaches.

## CCS Concepts

• **Applied computing → Online auctions**.

## Keywords

online bidding, Return-on-Spend, constrained bandits, UCB

**ACM Reference Format:**
Anonymous Author(s). 2018. Online Bidding under RoS Constraints without Knowing the Value. In *Proceedings of Make sure to enter the correct conference title from your rights confirmation emai (Conference acronym 'XX)*. ACM, New York, NY, USA, 12 pages. https://doi.org/XXXXXXX.XXXXXXX

## 1 Introduction

Online advertising, a multi-billion dollar industry, relies on real-time auctions to connect advertisers with users. These auctions, triggered by user queries or website visits, allow advertisers to bid for advertising slots, such as prominent placements on search engine results pages or in social media feeds. Advertisers aim to

maximize their returns, measured in conversions or other relevant metrics, by carefully determining their bids while adhering to budget constraints and desired return-on-spend (RoS) targets. To achieve this, a wide array of bidding strategies have been developed, leveraging techniques from optimization, online learning, and game theory to maximize advertiser utility [2, 7, 11, 23, 29, 30, 36, 47, 52].

Despite the sophistication of these strategies, many rely on the assumption that perfect knowledge of the value an impression generates is available to the advertiser beforehand. In reality, however, advertisers frequently face uncertainty about the true value of an ad impression, especially when dealing with new ad campaigns or evolving user preferences (cf. Section 2 for more details). In this work, we consider the practical scenario in which the value of an impression is *unknown* a priori and focus on developing bidding strategies that simultaneously learn the value of ad impressions as well as maximize the realized value of the advertiser.

Specifically, we study the problem of bidding for a single advertiser subject to total budget and RoS constraints. The budget constraint limits the total expenditure, while the RoS constraint ensures that the ratio of total value to total spend meets a predefined target, thus effectively capturing performance goals like target cost-per-acquisition (tCPA) and target return-on-ad-spend (tROAS), widely used in real-world advertising campaigns.[1]

We consider a stochastic setting where search queries and associated auctions arise dynamically. In this setting, the competing bids and the value of winning an auction are assumed to be sampled independently and identically distributed (i.i.d.) from an unknown distribution. In each round, the advertiser submits a bid without knowing the query's value beforehand. Upon bid submission, the auction mechanism determines the winner and price, with the value being revealed only if the advertiser wins. Our goal is to design an online bidding algorithm that maximizes the bidder's value over the entire horizon, while respecting the RoS and budget constraints.

### 1.1 Our Main Result

We evaluate our algorithm's performance via the notion of *regret* (Equation (2.5)), which quantifies the difference between its expected cumulative value and that achieved by an oracle, which possesses complete knowledge of the underlying competing bid and value distributions and employs a fixed strategy optimized for maximum cumulative value. Our main result now follows.

**Theorem** (Informal; see Theorem 3.3). *We propose an algorithm (Algorithm 3) designed for value maximization in online advertising auctions with return-on-spend (RoS) and budget constraints, without any prior knowledge of the values associated with incoming user queries. In the stochastic setting described earlier, with an online horizon of length $T$, our algorithm provably achieves $O(\sqrt{T \log(|\mathcal{B}|T)/V})$ regret for the objective of value maximization and $O(\sqrt{T \log(|\mathcal{B}|T)/V})$*

---

[1]See Google ads support page and Meta business help center for examples.

violation of the RoS constraint and $O(\sqrt{T \log(|\mathcal{B}|T)})$ violation of the budget constraint. Here, $\mathcal{B}$ is the domain of possible discrete bids and $V$ is the maximum per-round value achieved by the above oracle.

*Comparison to prior work.* To the best of our knowledge, ours is the first algorithm to achieve near-optimal regret and constraint violation bounds in this setting *without* knowledge of item values. This significantly extends recent work, which crucially assumes that the values are known to the bidder before bidding [19, 26, 38]. While [18] also addresses the setting with unknown values, their regret and constraint violation bounds incur a dependence of $O(\sqrt{|\mathcal{B}|})$, which we improve to a *logarithmic* dependence on $|\mathcal{B}|$. Additionally, [18] requires assuming the existence of a Slater point (a strictly feasible solution [17]), which limits the generality of their approach. Although [14] removes this assumption, it still incurs a $O(\sqrt{|\mathcal{B}|})$ dependence in the regret bounds, which is exponentially weaker than the logarithmic dependence achieved by our bound. In contrast to prior works, our work replaces the assumption of a Slater point with a milder condition that $V$ is bounded away from zero (see Section 1.2 for a detailed discussion). Furthermore, based on the lower bounds established by Achddou et al. [1] for utility maximization (without RoS constraints) under second-price auctions, we believe that a dependence on $V$ is unavoidable.

*Our core strengths.* A key strength of our algorithm and analysis is its simplicity. Unlike prior work that predominantly adopts a primal-dual approach — requiring intricate analysis of dual variables and assumptions like Slater's condition — we completely eliminate these restrictive (and often impractical) assumptions, making our approach both stronger and more general. Furthermore, by employing the upper confidence bound (UCB) framework, our method becomes easier to analyze (Sections 1.2 and 3) and simpler to implement, requiring very few hyper-parameters (cf. Section 4).

*Computational aspects.* A key challenge in our algorithm lies in efficiently estimating the arm recommended by the UCB. This estimation requires solving, in each round, a complex non-convex problem (Problem 3.5). We address this challenge by providing a computationally efficient technique that runs in $O(|\mathcal{B}|^3)$ time. This technique leverages the key insight that both the allocation and payment functions in standard auctions are monotonically increasing, which holds for both truthful auctions (e.g., second-price) and non-truthful auctions (e.g., first-price, all-pay) [34].

## 1.2 Key Technical Contributions

The problem of constrained reward-maximization may naturally be cast as one maximizing the "price-adjusted reward" (i.e., the reward minus a penalty on the constraint violation, with the penalty weighted by the dual variable associated with the constraint). This is the approach that has been widely adopted by much of the past work [10, 18, 26, 50]. The key idea is that as a constraint approaches violation, the corresponding dual variable grows large, signaling the need to bid conservatively in the next round; conversely, when previous bids create a buffer in the constraint, the dual variable shrinks, encouraging more aggressive bidding in the subsequent round.

However, in this primal-dual approach, one must assume the existence of a Slater point — an action which *strictly* satisfies the expected constraints. As dual variable magnitudes are bounded by

$O(1/\kappa)$ (cf. [13, Theorem 8.42]), where $\kappa$ is the minimum constraint slack of the Slater point, primal-dual methods can incur $O(1/\kappa)$ regret and constraint violation. In online bidding with RoS and budget constraints, bids with $\kappa \approx 0$ are common (e.g., when competing bids are narrowly concentrated). Consequently, algorithms based on the primal-dual approach will fundamentally incur large regret and constraint violation. We circumvent this shortcoming by introducing a UCB-style algorithm.

In a typical UCB-style algorithm (see, for example, [35, Chapter 7]), the idea is to create confidence sets for the unknown rewards and select the arm with the highest upper confidence bound. Our primary insight is to extend this principle to the constrained setting inherent in online bidding. In particular, we maintain appropriate confidence sets for both the constraints and rewards. Our algorithm then selects the bid that maximizes the reward while satisfying the constraints based on these confidence sets. This approach entirely eliminates the need for a Slater point (and, hence, avoids a regret dependence of $1/\kappa$) and addresses the problem even when the value is unknown. Finally, as noted earlier, finding this reward-maximizing bid is a highly nonconvex optimization problem. By utilizing the structure inherent to autobidding, we derive a provably efficient solution that is also easy to implement (Lemma 3.2).

## 1.3 Related Work

Our problem falls under the broader umbrella of bandit optimization under long-term constraints and has witnessed a long line of work by various research communities e.g. Agrawal and Devanur [3], Badanidiyuru et al. [9], Balseiro et al. [10], Castiglioni et al. [18], Gao et al. [29], Immorlica et al. [32], Mahdavi et al. [39, 40], Mannor et al. [41], Yu et al. [50], Yu and Neely [51].

Most of these works study the *budget/packing constraint*, e.g., Devanur et al. [24] obtain the optimal $O(\sqrt{T})$-regret under linear objective and constraints, Agrawal and Devanur [3] generalize it to nonlinear objectives, and Balseiro et al. [10] generalize it to nonlinear budget constraints. The RoS constraint we study differs fundamentally from the packing constraint studied in these works as well as in [9, 32]. There also exist papers that study a variant of our problem with a constraint class more general than ours (e.g., Agrawal and Devanur [3], Castiglioni et al. [18]); however, their guarantees for our problem are not as strong as ours, as we elaborate next.

For example, Castiglioni et al. [18] use a primal-dual framework for regret minimization with bandit feedback, which, when adapted to our bidding problem under the RoS constraint, achieves $\widetilde{O}(T^{3/4})$ regret with $\widetilde{O}(T^{3/4})$ constraint violation. Another crucial difference from our setting is that we do *not* know the values of the bids, whereas [18] (when adapted to this problem) does. Their bounds improve to our $\widetilde{O}(\sqrt{T})$ bounds under a 'strictly feasible' assumption; however, we require no such assumptions to get these bounds. In follow-up work, again with bandit feedback, Bernasconi et al. [14] study "Best of Both Worlds"-type algorithms for constrained regret minimization without the Slater point assumption. This work is closest to ours, but with the regret and constraint violation bounds suffering from an $O(\sqrt{|\mathcal{B}|})$ dependence (just like [18]), which can be substantial in practical settings; our work, in contrast, achieves an $O(\log(|\mathcal{B}|))$ dependence. This difference in the bidding setting arises because their observational model does not account for the

specifics of allocation and pricing functions. We empirically compare these algorithms against ours over synthetically generated bidding instances (cf. Section 4 for details).

Another example is the work of Agrawal and Devanur [3], which considers general online optimization with convex constraints. This work uses black-box low-regret methods with a strongly convex regularizer over the dual space. A sub-linear regret bound is attainable only when the dual space is well-bounded (e.g., a scaled simplex) or when the dual variable can be projected onto such a space without incurring too much additional regret. This canonical approach proves difficult for the RoS constraint, which can incur poor problem-specific parameters in generic guarantees. Hence, this technique cannot give sub-linear regret for the RoS constraint.

A recent line of work studies our bidding problem under both budget and RoS constraints; however, in each of these papers, the underlying assumption is that the bidder *knows* the value before submitting its bid. For example, Feng et al. [26] provide $\widetilde{O}(T^{1/2})$ regret and almost-sure constraint satisfaction using a primal-dual algorithm. The work of Lucier et al. [38], also in this setting, additionally obtains vanishing regret in the adversarial setting and provides aggregate guarantees on the resulting expected liquid welfare when multiple autobidders all deploy their algorithm. Other closely related works include those of Golrezaei et al. [30] and Celli et al. [19], the latter also considering multiple different constraints. However, as noted earlier, all of these require knowing the value.

The following works study regret minimization with the bidder *not* knowing the value. Weed et al. [49] study this for second-price auctions, and Achddou et al. [1] and Feng et al. [27] study this for general auctions, proving $O(\sqrt{T})$ regret bounds, with the former in a stochastic setting and the latter in an adversarial one. However, we note that all these works focus only on the unconstrained setting and maximize the utility, which is defined as the difference between the received value and the paid price.

A closely related line of work studies bandit optimization under long-term constraints [28, 37, 45, 53]. The works of Liu et al. [37] and Gangrade et al. [28] study this for linear bandits with long-term linear constraints. Liu et al. [37] use a primal-dual approach to provide $\widetilde{O}(d\sqrt{T})$ rates (where $d$ is the problem dimension), but require the existence of a Slater point. Gangrade et al. [28] avoid the need for Slater points, by maintaining doubly optimistic constraints and reward estimates, and obtain $\widetilde{O}(d\sqrt{T})$ rates. Both these works, when specialized to autobidding, incur linear dependence on $|\mathcal{B}|$ in the regret. This problem was also studied for kernelized bandits in [53], but their algorithm requires knowledge of a lower bound on the Slater slack. A different, but related, problem of satisfying constraints in each round is studied in [45]. This work shows that knowledge of a "safe" action is necessary for per-round constraint satisfaction and obtains $O(\sqrt{T})$ regret under this assumption.

Finally, the related problem of learning to bid in repeated auctions has been explored in both academia and industry, e.g. Badanidiyuru et al. [8], Borgs et al. [16], Feng et al. [27], Han et al. [31], Nedelec et al. [42], Noti and Syrgkanis [44], Weed et al. [49]. These works abstract the problem of learning to bid as one of contextual bandits, but do not incorporate constraints into them. Beyond these, there has been some work on bidding under budget constraints, e.g.,

Ai et al. [4], Balseiro and Gur [12]. However, these papers focus on utility-maximizing agents with at most one constraint.

## 2 Preliminaries

We consider an auction with multiple bidders and study the online bidding problem from the perspective of a single learner (bidder). At each time step $t$, nature generates an ad query associated with a value $v_t \in [0, 1]$ and an auction mechanism $(x_t, p_t)$. The auction mechanism is determined by the allocation and payment functions:

- *Allocation function,* $x_t : \mathcal{B} \to [0, 1]$, which specifies the probability of winning the auction for a given bid. We define $x_t(\,\cdot\,) :=$ $x(\,\cdot\,, B_t^{\complement})$, where $\mathcal{B}$ is a finite subset of $\mathbb{R}_{\geq 0}$ with $0 \in \mathcal{B}$ (i.e., the bidder can submit a bid of zero), and $B_t^{\complement}$ denotes the vector of bids of the other bidders at time step $t$. Observe that the allocation probability depends not only on the learner's bid but also on the bids of other participants.

- *Payment function,* $p_t : \mathcal{B} \to [0, 1]$, which determines the payment required when the auction is won. Similar to the allocation function, we define $p_t(\,\cdot\,) := p(\,\cdot\,, B_t^{\complement})$. We assume that the payment is zero when the allocation is zero and is always at most the submitted bid. This ensures that the bidder never pays more than their bid, a standard assumption in auctions.

We use the shorthand $q_t(b) := x_t(b) \cdot p_t(b)$ to denote the price paid for a bid $b$. A key distinction of our model from those in prior works [1, 26] is that we do not assume the auctions to be truthful. Instead, all we require is that the functions $x_t(\,\cdot\,)$ and $p_t(\,\cdot\,)$ be *monotonic*, a property satisfied by many popular auctions, including first-price, second-price, and all-pay auctions [33].

Another important point of departure from previous work is that, at each time step $t$, the value $v_t$ is *unknown* to the learner before submitting a bid. This model reflects the uncertainty inherent in many online advertising scenarios. The learner decides its bid $b_t$ based on all the information obtained so far. After submitting its bid, the learner observes the outcome from the auction mechanism, i.e., $x_t(\cdot)$ and $p_t(\cdot)$. If the bidder wins, then the auction mechanism also reveals the value $v_t$. For a bid $b$ with value $v$, allocation function $x(\,\cdot\,)$, and payment function $p(\,\cdot\,)$, the *realized* value and *paid* price are $v \cdot x(b)$ and $v \cdot p(b)$, respectively.

This setting of unknown value is common in several practical online bidding environments. Examples include advertisers who participate infrequently in auctions, new ad campaigns with uncertain performance, or scenarios where the value of an advertisement is influenced by multiple factors, such as clicks, conversions, brand awareness, and customer lifetime value. Even in autobidding systems [22], where machine learning models predict clicks and conversions to inform bidding algorithms, these predictions often capture only partial information about the true value and can be inaccurate, especially for new or infrequently shown advertisements.

Similar to prior works on online bidding [14, 18, 46], we assume a stochastic setting where the auction environment is governed by an underlying probability distribution. Specifically, for all $t \in [T]$, the tuple $\gamma_t := (v_t, x_t, p_t)$ is drawn independently and identically (i.i.d.) from an unknown distribution $\mathcal{P}$. This implies that the sequence of $T$ samples, denoted by $\overrightarrow{\gamma} := \{\gamma_1, \gamma_2, \ldots, \gamma_T\}$, follows the product distribution $\mathcal{P}^T$. This induces the expectations $\bar{v} := \mathbb{E}[v_t]$, $\bar{q}(b) :=$ $\mathbb{E}[x_t(b)p_t(b)]$, and $\bar{x}(b) := \mathbb{E}[x_t(b)]$ for any bid $b \in \mathcal{B}$.

We design online bidding algorithms to maximize the learner's total realized value subject to RoS and budget constraints. Formally, this optimization problem is given by

$$
\begin{aligned}
\underset{b_t : t=1,\cdots,T}{\text{maximize}} \quad & \sum_{t=1}^{T} v_t \cdot x_t(b_t) \\
\text{subject to} \quad & \text{RoS} \cdot \sum_{t=1}^{T} q_t(b_t) \leq \sum_{t=1}^{T} v_t \cdot x_t(b_t), \\
& \sum_{t=1}^{T} q_t(b_t) \leq \rho T,
\end{aligned}
\tag{2.1}
$$

where $\text{RoS} > 0$ is the target ratio of the RoS bidder and $\rho T$ the total budget, with $\rho > 0$ (assumed a fixed constant) measuring the limit of the average expenditure over $T$ rounds (ad queries). Throughout the paper we assume without loss of generality[2] that $\text{RoS} = 1$.

*Analysis setup.* We use the notions of regret and constraint violation to measure the performance of our algorithm. To define regret, we first define the reward collected by our algorithm ("Alg") for a sequence of requests $\overrightarrow{\gamma}$ over a time horizon $T$ as

$$
\text{Reward}(\text{Alg}, \overrightarrow{\gamma}) \coloneqq \sum_{t=1}^{T} v_t \cdot x_t(b_t).
\tag{2.2}
$$

To define the benchmark against which we measure the regret of Alg, we consider the following linear program (LP):

$$
\begin{aligned}
\underset{w \in \Delta_{|\mathcal{B}|}}{\text{maximize}} \quad & \sum_{b \in \mathcal{B}} w(b) \cdot \overline{v} \cdot \overline{x}(b) \\
\text{subject to} \quad & \sum_{b \in \mathcal{B}} w(b) \cdot \overline{q}(b) \leq \sum_{b \in \mathcal{B}} w(b) \cdot \overline{v} \cdot \overline{x}(b), \\
& \sum_{b \in \mathcal{B}} w(b) \cdot \overline{q}(b) \leq \rho.
\end{aligned}
\tag{2.3}
$$

and let us denote the value of this LP as $V$ and its optimizer as $w^*_{\text{LP}}$. Here $\Delta_{|\mathcal{B}|}$ is the set of all probability distributions over $\mathcal{B}$. We define our benchmark to be:

$$
\text{Reward}(\text{Opt}) \coloneqq \sum_{t=1}^{T} \sum_{b \in \mathcal{B}} w^*_{\text{LP}}(b) \cdot \overline{v} \cdot \overline{x}(b) = T \cdot V.
\tag{2.4}
$$

Thus, we are comparing against an algorithm that has knowledge of $\overline{v}$, $\overline{x}$, and $\overline{q}$, and plays a bid sampled from $w^*_{\text{LP}}$ for each of the $T$ rounds. This is a commonly used benchmark in the stochastic setting [14, 18, 46]. These definitions lead to the following definition of regret of Alg in this setup:

$$
\text{Regret}(\text{Alg}, \mathcal{P}^T) \coloneqq \text{Reward}(\text{Opt}) - \mathbb{E}_{\overrightarrow{\gamma} \sim \mathcal{P}^T} \left[ \text{Reward}(\text{Alg}, \overrightarrow{\gamma}) \right].
\tag{2.5}
$$

We remark that Reward is defined for some specific input sequence, whereas Regret is defined with respect to a distribution. Additionally, we define budget and RoS constraint violations as $\sum_{t=1}^{T} q_t(b_t) - \rho T$ and $\sum_{t=1}^{T} (q_t(b_t) - v_t \cdot x_t(b_t))$, respectively.

## 3 UCB-RoS

In this section, we solve the online bidding problem formalized in Problem 2.1 by designing a novel UCB-style algorithm (presented in Algorithm 3). Our approach draws inspiration from the UCB technique widely used in the bandit literature [5].

We rely on the principle of "optimism in the face of uncertainty." At each time step, our algorithm maintains confidence sets for the unknown parameters of the problem, namely the allocation function, the pricing function, and the value distribution. It then selects the bid that maximizes the expected reward within these confidence sets. These confidence intervals are carefully designed

to satisfy two key properties: (1) they contain the true expected values with high probability, and (2) they shrink as more data is collected, reflecting increasing confidence in the estimates.

As mentioned earlier, finding the reward-maximizing bid within these confidence intervals is challenging. This optimization problem is inherently non-convex and can be computationally intractable in general. However, by exploiting the specific structure of typical auctions, we derive a simple and efficient solution to this problem, as detailed in Lemma 3.2. We now expand upon these ideas.

Recalling our setup, after submitting bid $b_t$, the bidder obtains the allocation $x_t(\cdot)$ and price function $p_t(\cdot)$. Additionally, if the bid is won, then it also obtains its value $v_t$. Let $N_t$ denote the number of times the user wins the bid in the first $t$ rounds. Then, the algorithm at time step $t$ updates its sample estimators for the allocation, pricing functions, and value in the following way:

$$
\widehat{x}_t(\cdot) \coloneqq \sum_{s=1}^{t} \frac{x_s(\cdot)}{t}, \quad \widehat{q}_t(\cdot) \coloneqq \sum_{s=1}^{t} \frac{x_s(\cdot) p_s(\cdot)}{t}, \quad \widehat{v}_t \coloneqq \sum_{s=1}^{N_t} \frac{v_s}{N_t}.
\tag{3.1}
$$

We remark that the first two estimators are functions defined on $\mathcal{B}$ and taking values in $[0, 1]$, while the value estimator takes real values built only from the subset of samples in which the algorithm wins the bid. Next, we describe our construction of confidence intervals around these estimators, in which we later show (Lemma 3.1) the true expected quantities lie with high probability.

*Constructing confidence sets.* For every $t \in [T]$, the algorithm constructs confidence sets centered around the sample estimators defined in Equation (3.1). To introduce these constructions, we first let $\mathcal{M}$ denote the set of all non-decreasing functions $f$ on $\mathcal{B}$, taking values in $[0, 1]$. Then, these confidence sets are defined as:

$$
C^{\widehat{x}_t} \coloneqq \left\{ f \in \mathcal{M} \mid |f(b) - \widehat{x}_t(b)| \leq \sqrt{\frac{\log(2|\mathcal{B}|T)}{2t}}, \forall b \in \mathcal{B} \right\},
$$

$$
C^{\widehat{q}_t} \coloneqq \left\{ f \in \mathcal{M} \mid |f(b) - \widehat{q}_t(b)| \leq \sqrt{\frac{\log(2|\mathcal{B}|T)}{2t}}, \forall b \in \mathcal{B} \right\},
$$

$$
C^{\widehat{v}_t} \coloneqq \left\{ v \in [0, 1] \mid |v - \widehat{v}_t| \leq \sqrt{\frac{\log(2T)}{2N_t}} \right\}.
\tag{3.2}
$$

Interestingly, the confidence set $C^{\widehat{q}_t}$ does not require its constituent functions to be of the form $x \cdot p$, rather only that they are clustered around $\widehat{q}$. This generality proves crucial in Lemma 3.2. These confidence sets have been constructed to ensure that for all bids, the expectations of the true allocation functions $\overline{x}(\cdot)$, pricing functions $\overline{q}(\cdot)$, and values $\overline{v}$ fall, with high probability, within their respective confidence sets. More precisely, we have the following.

LEMMA 3.1. *With probability at least* $1 - \frac{1}{T}$*, for every* $t \in [T]$ *it holds that* $\overline{x}(\cdot) \in C^{\widehat{x}_t}$, $\overline{q}(\cdot) \in C^{\widehat{q}_t}$, *and* $\overline{v} \in C^{\widehat{v}_t}$, *where* $C^{\widehat{x}_t}$, $C^{\widehat{q}_t}$, *and* $C^{\widehat{v}_t}$ *are as defined in Equation* (3.2).

PROOF. As a result of the imposed ranges on $x_t$ and $p_t$, we infer that for each bid $b \in \mathcal{B}$, the allocation function $x_t(\cdot)$ and the pricing function $q_t(\cdot)$ satisfy the bounded difference property, i.e., for any $(x_t, p_t)$ and $(x'_t, p'_t)$,

$$
|x_t(b) - x'_t(b)| \leq 1
\tag{3.3}
$$

---

[2]For any RoS $\neq 1$, we can scale the values to be $v_t \coloneqq \text{RoS} \cdot v_t$.

and that

$$|x_t(b) \cdot p_t(b) - x_t'(b) \cdot p_t'(b)| \leq 1. \qquad (3.4)$$

Since we assume stochastic behaviour for $(x_t, p_t)$, we can apply Hoeffding inequality (Fact A.1) on the estimators $\widehat{x}_t(\,\cdot\,)$, $\widehat{q}_t(\,\cdot\,)$ and $\widehat{v}_t$ to get the following probabilities for each $b \in \mathcal{B}$:

$$\mathbb{P}\left( \left| \widehat{x}_t(b) - \overline{x}(b) \right| > \sqrt{\frac{1}{2t} \log\left(\frac{1}{\delta_1}\right)} \right) \leq \delta_1,$$

$$\mathbb{P}\left( \left| \widehat{q}_t(b) - \overline{q}(b) \right| > \sqrt{\frac{1}{2t} \log\left(\frac{1}{\delta_2}\right)} \right) \leq \delta_2,$$

$$\mathbb{P}\left( \left| \widehat{v}_t - \overline{v} \right| > \sqrt{\frac{1}{2N_t} \log\left(\frac{1}{\delta_3}\right)} \right) \leq \delta_3.$$

Choosing $\delta_1 = \delta_2 = \frac{1}{2|\mathcal{B}|T}$ and $\delta_3 = \frac{1}{2T}$ and taking a union bound over $b \in \mathcal{B}$ and $t \in [T]$ gives the result. $\qquad \square$

*Updating the bid.* Having updated the sample estimators and confidence sets according to (3.1) and (3.2), the algorithm computes the bid for the next round by maximizing the bidder's realized value, while also meeting the per round expected constraint satisfaction. Specifically, the algorithm solves the optimization problem:

$$
\begin{aligned}
\underset{w, x, q, v}{\text{maximize}} \quad & \sum_{b \in \mathcal{B}} w(b) \cdot v \cdot x(b) \\
\text{subject to} \quad & w \in \Delta_{|\mathcal{B}|}, \ x \in C^{\widehat{x}_t}, \ q \in C^{\widehat{q}_t}, \ v \in C^{\widehat{v}_t} \\
& \sum_{b \in \mathcal{B}} w(b) \cdot q(b) \leq \sum_{b \in \mathcal{B}} w(b) \cdot v \cdot x(b), \\
& \sum_{b \in \mathcal{B}} w(b) \cdot q(b) \leq \rho.
\end{aligned} \qquad (3.5)
$$

This formulation essentially instantiates our original problem (Problem 2.1), with the price and value estimates drawn from the updated confidence sets computed thus far. The algorithm samples a bid $b_{t+1} \sim w_{t+1}^*$, where $w_{t+1}^*$ is the optimal distribution obtained from Problem 3.5 and submits this updated bid in the next round $(t + 1)$. It is important to note that in Problem 3.5, the variables of optimization are $w(\,\cdot\,)$, $x(\,\cdot\,)$, $q(\,\cdot\,)$, and $v$, with the confidence sets $C^{\widehat{x}_t}$, $C^{\widehat{q}_t}$, and $C^{\widehat{v}_t}$ being the only known quantities. Therefore, Problem 3.5 is highly nonconvex, and a priori, it is unclear how to solve it. However, through the use of specific structure in our problem, we show (Lemma 3.2) that this can indeed be done efficiently.

*Computational complexity.* For Problem 3.5 derived from our online bidding setup, the optimizers $x_t^*$, $q_t^*$, $v_t^*$, and $w_t^*$ can be found with at most $O(|\mathcal{B}|^3)$ computational steps, as we elaborate next. The following lemma gives the explicit form of the optimizers.

LEMMA 3.2. *Let $x_t^*$, $v_t^*$, $q_t^*$, and $w_t^*$ be the optimizers of Problem 3.5. Let $N_t$ be the number of times the bidder wins the bid until time $t$. Then, the optimizers can be explicitly written as follows.*

$$x_t^*(b) = \min\left\{ \widehat{x}_t(b) + \sqrt{\frac{1}{2t} \log(2|\mathcal{B}|T)}, 1 \right\}$$

$$v_t^* = \min\left\{ 1, \widehat{v}_t + \sqrt{\frac{1}{N_t} \log(2T)} \right\}$$

$$q_t^*(b) = \max\left\{ 0, \widehat{q}_t(b) - \sqrt{\frac{1}{2t} \log(2|\mathcal{B}|T)} \right\},$$

---

**Algorithm 3.1** UCB-RoS

**Input:** bid set $\mathcal{B}$, per-round budget $\rho$, bidding horizon $T$.
**Initialize:** $t \leftarrow 1$, $b_1 \leftarrow \max_{b \in \mathcal{B}} \{b\}$, $C^{\widehat{v}_0} \leftarrow [0, 1]$, $N_1 \leftarrow 0$, $\widehat{v}_1 \leftarrow 0$.

1: Submit bid $b_1$ and observe $x_1(\,\cdot\,)$ and $p_1(\,\cdot\,)$.
2: Set $\widehat{x}_1(\,\cdot\,) \leftarrow x_1(\,\cdot\,)$ and $\widehat{q}_1(\,\cdot\,) \leftarrow x_1(\,\cdot\,)p_1(\,\cdot\,)$.
3: Set $C^{\widehat{v}_1} \leftarrow C^{\widehat{v}_0}$
4: **if** the bidder wins **then**
5:     Observe value $v_1$.
6:     Set $N_1 \leftarrow 1, \widehat{v}_1 \leftarrow v_1$.
7: **end if**
8: Update $C^{\widehat{x}_1}, C^{\widehat{q}_1}, C^{\widehat{v}_1}$ using Equation (3.2).
9: **for** $t = 2$ to $T$ **do**
10:     Compute $x_t^*, q_t^*, v_t^*,$ and $w_t^*$, the optimizers of Problem 3.5 defined by the confidence sets $C^{\widehat{x}_{t-1}}, C^{\widehat{q}_{t-1}},$ and $C^{\widehat{v}_{t-1}}$.
11:     Sample $b_t \sim w_t^*$.
12:     Submit bid $b_t$ and observe $x_t(\,\cdot\,)$ and $p_t(\,\cdot\,)$.
13:     Update the allocation and pricing function estimates:

$$\widehat{x}_t(\,\cdot\,) \leftarrow \frac{(t-1)\widehat{x}_{t-1}(\,\cdot\,) + x_t(\,\cdot\,)}{t},$$

$$\widehat{q}_t(\,\cdot\,) \leftarrow \frac{(t-1)\widehat{q}_{t-1}(\,\cdot\,) + q_t(\,\cdot\,)}{t}.$$

14:     Set $\widehat{v}_t \leftarrow \widehat{v}_{t-1}$
15:     **if** the bidder wins **then**
16:         Observe value $v_t$.
17:         Update $N_t \leftarrow N_{t-1} + 1, \quad \widehat{v}_t \leftarrow \frac{(N_t - 1)\widehat{v}_{t-1} + v_t}{N_t}$.
18:     **end if**
19:     Update $C^{\widehat{x}_t}, C^{\widehat{q}_t}, C^{\widehat{v}_t}$ using Equation (3.2).
20: **end for**

---

*where $\widehat{x}_t$, $\widehat{v}_t$, and $\widehat{q}_t$ are defined in Equation (3.1). Finally, $w^*(b)$ may be computed in terms of the above quantities as*

$$
\begin{aligned}
w_t^* = \underset{w \in \Delta_{|\mathcal{B}|}}{\text{argmax}} \quad & \sum_{b \in \mathcal{B}} w(b) \cdot v_t^* \cdot x_t^*(b) \\
\text{subject to} \quad & \sum_{b \in \mathcal{B}} w(b) \cdot q_t^*(b) \leq \sum_{b \in \mathcal{B}} w(b) \cdot v_t^* \cdot x_t^*(b) \\
& \sum_{b \in \mathcal{B}} w(b) \cdot q_t^*(b) \leq \rho.
\end{aligned}
$$

PROOF. By construction, $C^{\widehat{q}_t}$ and $C^{\widehat{x}_t}$ are sets of monotone functions on $\mathcal{B}$. For any choice of allocation, pricing functions, and value from the confidence sets in Equation (3.2), define the set

$$S(x, q, v) = \left\{ w \in \Delta_{\mathcal{B}} \mid \sum_{b \in \mathcal{B}} w(b) \cdot q(b) \leq \min\left( \rho, \sum_{b \in \mathcal{B}} w(b) \cdot v \cdot x(b) \right) \right\}$$

and the value obtained by the following maximization:

$$
\begin{aligned}
f(x, q, v) = \underset{w \in \Delta_{|\mathcal{B}|}}{\text{maximize}} \quad & \sum_{b \in \mathcal{B}} w(b) \cdot v \cdot x(b) \\
\text{subject to} \quad & \sum_{b \in \mathcal{B}} w(b) \cdot q(b) \leq \sum_{b \in \mathcal{B}} w(b) \cdot v \cdot x(b) \\
& \sum_{b \in \mathcal{B}} w(b) \cdot q(b) \leq \rho.
\end{aligned}
$$

Consider two functions $x^0(\,\cdot\,)$ and $x^1(\,\cdot\,)$ such that:

$$x^1(b) \leq x^0(b), \ \forall b \in \mathcal{B}.$$

This then implies that for any *fixed* choice of $q$ and $v$, we have

$$\sum_{b \in \mathcal{B}} w(b) \cdot v \cdot x^1(b) \leq \sum_{b \in \mathcal{B}} w(b) \cdot v \cdot x^0(b), \quad \forall b \in \mathcal{B} \tag{3.6}$$

$$S(x^1, q, v) \subseteq S(x^0, q, v).$$

Then, combining (3.6) with the definition of $f$ implies that

$$f(x^1, q, v) \leq f(x^0, q, v).$$

We can then infer that, for any fixed choice of $v$ and $q$, the maximizer of Problem 3.5 is the function that chooses the upper confidence bound of the current confidence set $C^{\widehat{x}_t}$, i.e.,:

$$x_t^*(\,\cdot\,) = \min\left\{1, \widehat{x}_t(\,\cdot\,) + \sqrt{\frac{1}{2t} \log(2|\mathcal{B}|T)}\right\}$$

An analogous argument can be applied to show that $v_t^*$ is:

$$v_t^* = \min\left\{1, \widehat{v}_t + \sqrt{\frac{1}{2N_t} \log(2T)}\right\}$$

and that $q_t^*$ is given by the lower confidence bound of the set $C^{\widehat{q}_t}$:

$$q_t^*(\,\cdot\,) = \max\left\{0, \widehat{q}_t(\,\cdot\,) - \sqrt{\frac{1}{2t} \log(2|\mathcal{B}|T)}\right\}.$$

The form of $w_t^*$ is obtained by plugging back into Problem 3.5 the explicit form of $x_t^*$, $q_t^*$, and $v_t^*$ obtained above. □

Since $w_t^*$ is a solution to an LP, one can explicitly compute it with at most $O(|\mathcal{B}|^3)$ computational effort [21].

*Regret and constraint violation bound.* Our main result below guarantees an $\widetilde{O}(\sqrt{T})$ regret and constraint violation bound. We call the event when the concentration results in Lemma 3.1 and Lemma B.1 hold as *clean execution* and note that it occurs with probability at least $1 - \frac{3}{T}$.

**Theorem 3.3.** *Consider the online bidding problem described in Section 2. Let $V$ be the value of the LP defined in Equation (2.3). For any time horizon $T$, Algorithm 3 suffers the following regret bound in expectation:*

$$\mathbb{E}\left[\text{Regret}(\text{Alg}, \mathcal{P}^T)\right] = O\left(\max\left\{\sqrt{\frac{T \log(|\mathcal{B}|T)}{V}}, \frac{\log(|\mathcal{B}|T)}{V^2}\right\}\right),$$

*where regret is as defined in Equation (2.5). Further, the violation of the RoS and budget constraint is, in expectation, at most $O\left(\sqrt{\frac{T \log(|\mathcal{B}|T)}{V}}\right)$ and $O(\sqrt{T \log(|\mathcal{B}|T)})$, respectively.*

PROOF. First, observe that under clean execution, we have

$$|x_t^*(b) - \overline{x}(b)| \leq \sqrt{\frac{2}{t} \log(2|\mathcal{B}|T)}.$$

Combining this result with Lemma B.1, and utilizing the fact that $w_s^*$ is a probability distribution over bids, gives us

$$\left|N_t - \sum_{s=1}^t \sum_{b \in \mathcal{B}} w_s^*(b) x_s^*(b)\right| \leq \frac{2 \log(T)}{V} + \frac{V \cdot t}{2} + \sum_{s=1}^t \sqrt{\frac{2 \log(2|\mathcal{B}|T)}{s}}. \tag{3.7}$$

Under clean execution, $\overline{x}(\,\cdot\,) \in C^{\widehat{x}_t}$, $\overline{q}(\,\cdot\,) \in C^{\widehat{q}_t}$, $\overline{v} \in C^{\widehat{v}_t}$ for each $t$ and hence $(\overline{x}(\,\cdot\,), \overline{q}(\,\cdot\,), \overline{v}, w_{\text{LP}}^*)$ is a feasible point for Problem 3.5.

Combining this observation with the optimality of $(x_s^*, q_s^*, v_s^*, w_s^*)$, we get that

$$V \leq \sum_{b \in \mathcal{B}} w_s^*(b) x_s^*(b) v_s^*$$

for each $s \leq t$. Noting that $v_s^* \leq 1$, and summing over $s$, we have that $t \cdot V \leq \sum_{s=1}^t \sum_{b \in \mathcal{B}} w_s^*(b) x_s^*(b)$. Using this in (3.7), we get:

$$N_t \geq \frac{t \cdot V}{2} - \frac{2 \log(T)}{V} - \sqrt{t \log(|\mathcal{B}T|)}. \tag{3.8}$$

Hence,

$$N_t \geq \frac{V \cdot t}{3}, \quad \forall t \geq \frac{24 \log(|\mathcal{B}|T)}{V^2}. \tag{3.9}$$

Consider the "per-round regret"

$$r_t := \sum_{b \in \mathcal{B}} w_{\text{LP}}^*(b) \cdot \overline{v} \cdot \overline{x}(b) - \sum_{b \in \mathcal{B}} w_t^*(b) \cdot \overline{v} \cdot \overline{x}(b).$$

We then have:

$$r_t \leq \sum_{b \in \mathcal{B}} w_t^*(b) \cdot v_t^* \cdot x_t^*(b) - \sum_{b \in \mathcal{B}} w_t^*(b) \cdot \overline{v} \cdot \overline{x}(b)$$

$$= \sum_{b \in \mathcal{B}} w_t^*(b) \cdot \left[v_t^* \cdot (x_t^*(b) - \overline{x}(b)) + \overline{x}(b)(v_t^* - \overline{v})\right]$$

$$\leq \sqrt{\frac{1}{2t} \log(2|\mathcal{B}|T)} + \sqrt{\frac{3}{2tV} \log(2|\mathcal{B}|T)} \tag{3.10}$$

$$\leq O\left(\sqrt{\frac{1}{tV} \log(|\mathcal{B}|T)}\right),$$

where the first step is by clean execution, Lemma 3.1, and the optimality, for Problem 3.5, of $v_t^*$, $x_t^*$, and $w_t^*$, all of which lie in the confidence intervals given by Lemma 3.1. In the third step we utilize the (3.9) and Lemma 3.1. Next, by definition of $r_t$, we note that $\sum_{t=1}^T r_t$ may equivalently be expressed as below:

$$\sum_{t=1}^T r_t = \text{Reward}(\text{Opt}) - \sum_{t=1}^T \mathbb{E}_{t-1}[v_t \cdot x_t(b_t)],$$

where $\mathbb{E}_t[.]$ is the conditional expectation. Computing the expectation over the randomness in the entire sequence of inputs gives:

$$\mathbb{E}_{\overrightarrow{\gamma} \sim \mathcal{P}^T}\left[\sum_{t=1}^T r_t\right] = \text{Reward}(\text{Opt}) - \mathbb{E}_{\overrightarrow{\gamma} \sim \mathcal{P}^T}\left[\text{Reward}(\text{Alg}, \overrightarrow{\gamma})\right]$$

$$= \text{Regret}(\text{Alg}, \mathcal{P}^T).$$

Because we have a bound on $\sum_{t=1}^T r_t$ under clean execution, which holds with a probability at least $1 - \frac{1}{T}$, we can bound the regret as:

$$\text{Regret}(\text{Alg}, \mathcal{P}^T) \leq \sum_{t=1}^T O\left(\sqrt{\frac{1}{Vt} \log(|\mathcal{B}|T)}\right)\left(1 - \frac{3}{T}\right) + 2T \cdot \frac{3}{T}$$

$$\leq O\left(\sqrt{\frac{T \log(|\mathcal{B}|T)}{V}}\right).$$

This completes the proof of the regret bound. We now proceed to bound the violation of the budget constraint under clean execution. To this end, we consider the following expression:

$$\sum_{b \in \mathcal{B}} w_t^*(b) \cdot \overline{q}(b) = \sum_{b \in \mathcal{B}} w_t^*(b) \cdot (\overline{q}(b) - q_t^*(b)) + \sum_{b \in \mathcal{B}} w_t^*(b) \cdot q_t^*(b). \tag{3.11}$$

By clean execution and Lemma 3.1, we have

$$\sum_{t=1}^{T}\sum_{b\in\mathcal{B}} w_t^*(b)\cdot(\overline{q}(b)-q_t^*(b)) \le \sum_{t=1}^{T}\sqrt{\frac{\log(2|\mathcal{B}|T)}{2t}}$$

$$\le O\left(\sqrt{T\log(|\mathcal{B}|T)}\right). \qquad (3.12)$$

Next, since $q_t^*$ satisfies the per round constraint, we have

$$\sum_{t=1}^{T}\sum_{b\in\mathcal{B}} w_t^*(b)\cdot q_t^*(b) \le \rho T. \qquad (3.13)$$

Plugging Inequalities (3.12) and (3.13) into Equation (3.11) yields

$$\sum_{t=1}^{T}\sum_{b\in\mathcal{B}} w_t^*(b)\cdot\overline{q}(b) \le O(\sqrt{T\log(|\mathcal{B}|T)}) + \rho T. \qquad (3.14)$$

The expression on the left-hand side of Inequality (3.14) may be expressed as $\sum_{t=1}^{T}\sum_{b\in\mathcal{B}} w_t^*(b)\cdot\overline{q}(b) = \sum_{t=1}^{T}\mathbb{E}_{t-1}[q_t(b_t)]$. The expected budget violation may then be bounded as follows:

$$\mathbb{E}_{\overrightarrow{\gamma}\sim\mathcal{P}^T}\left[\sum_{t=1}^{T} q_t(b_t) - \rho T\right] \le O(\sqrt{T\log(|\mathcal{B}|T)})\left(1-\frac{3}{T}\right) + \frac{3\rho T}{T}$$

$$\le O(\sqrt{T\log(|\mathcal{B}|T)}).$$

This concludes the proof of the bound on the total budget violation. To prove our bound on the RoS constraint violation, we apply a similar analysis, which we state here for completeness. Consider again Equation (3.11). Then, Inequality (3.12) holds again, due to clean execution. Continuing the analysis, we have

$$\sum_{t=1}^{T}\sum_{b\in\mathcal{B}} w_t^*(b)q_t^*(b) \le \sum_{t=1}^{T}\sum_{b\in\mathcal{B}} w_t^*(b)x_t^*(b)v_t^*,$$

because of optimality of $v_t^*$, $w_t^*$, $x_t^*$, and $q_t^*$ for Problem 3.5 (from Lemma 3.2). Repeating, on the right-hand side above, the steps from Inequality (3.10), we get the following bound:

$$\sum_{t=1}^{T}\sum_{b\in\mathcal{B}} w_t^*(b)\cdot q_t^*(b)$$

$$\le O\left(\sqrt{\frac{T\log(|\mathcal{B}|T)}{V}}\right) + \sum_{t=1}^{T}\sum_{b\in\mathcal{B}} w_t^*(b)\cdot\overline{v}\cdot\overline{x}(b). \qquad (3.15)$$

From Inequality (3.15), we can obtain the following bound:

$$\mathbb{E}_{\overrightarrow{\gamma}\sim\mathcal{P}^T}\left[\sum_{t=1}^{T} q_t(b_t) - \sum_{t=1}^{T} v_t\cdot x_t(b_t)\right]$$

$$\le O\left(\sqrt{\frac{T\log(|\mathcal{B}|T)}{V}}\right)\left(1-\frac{3}{T}\right) + \frac{6T}{T}$$

$$\le O\left(\sqrt{\frac{T\log(|\mathcal{B}|T)}{V}}\right).$$

This concludes the proof of the RoS constraint violation bound in expectation and therefore finishes the proof of the lemma. □

Observe that we have exhibit a logarithmic dependence on $|\mathcal{B}|$. This is in contrast with existing algorithms for this problem, which suffer from a $\sqrt{|\mathcal{B}|}$ dependence. Moreover, ignoring the dependence

on $\mathcal{B}$, our algorithm achieves $\widetilde{O}(\sqrt{T/V})$ regret and constraint violation bounds. In contrast, primal-dual approaches yield $\widetilde{O}(\sqrt{T/\kappa})$ bounds (where $\kappa$ is the Slater slack) [18]. In many practical scenarios, $\kappa$ can be very close to zero, while $V$ remains bounded away from zero (see Appendix D). Consequently, our approach provides significantly stronger guarantees in cases where primal-dual methods may suffer from large regret and constraint violations. Furthermore, we believe that a dependence on $V$ is unavoidable. This conjecture is supported by the lower bound established in [1] for online bidding with unknown value (albeit without RoS constraints), which depends on a quantity proportional to $1/V$.

*Remark* 3.1 (Extension to linear bandits). While our focus is on online bidding, our algorithm and analysis can be extended to the more general setting of stochastic linear bandits with linear long-term stochastic constraints (see Appendix E for results). The regret and constraint violation bounds in this case avoid the Slater slack $\kappa$, thus improving on the existing primal-dual algorithms.

We now strengthen the in-expectation regret and constraint violation bounds of Theorem 3.3 by providing high-probability guarantees. These stronger bounds are obtained by leveraging Azuma's inequality (Fact A.3) to bound the deviations of the key quantities from their expectations

**Theorem 3.4** (High Probability bounds). *Given i.i.d. inputs from a distribution $\mathcal{P}$ over a time horizon $T$ to Algorithm 3, with a probability at least $1-\frac{5}{T}$, we have that:*

$$\text{Reward}(\text{Opt}) - \text{Reward}(\text{Alg},\overrightarrow{\gamma}) \le O\left(\sqrt{\frac{T\log(|\mathcal{B}|T)}{V}}\right),$$

$$\sum_{t=1}^{T} q_t(b_t) \le \rho T + O(\sqrt{T\log(|\mathcal{B}|T)}),$$

$$\sum_{t=1}^{T} q_t(b_t) \le \sum_{t=1}^{T} v_t\cdot x_t(b_t) + O\left(\sqrt{\frac{T\log(|\mathcal{B}|T)}{V}}\right).$$

Theorem 3.4 demonstrates that, with high probability, the sample pathwise constraint violation of Algorithm 3 is bounded by $O\left(\sqrt{\frac{T\log(|\mathcal{B}|T)}{V}}\right)$. This strengthens the in-expectation bounds from Theorem 3.3 (see Appendix C for the proof).

## 4 Experiments

We empirically study the performance of UCB-RoS and compare it with other existing approaches on synthetically generated datasets. We create synthetic problems where $v_t$, $x_t(\cdot)$, $q_t(\cdot)$, and $B_t^{\mathsf{C}}$ are sampled i.i.d. from specified distributions. Given a pre-specified mean value $\overline{v}_t$, the values $v_t$ are sampled i.i.d. from a corresponding beta distribution with shape parameters $(10\overline{v}, 10\cdot(1-\overline{v}))$. The bidding set $\mathcal{B}$ is assumed to be a uniformly spaced grid over $[0,1]$ with grid size of $1/|\mathcal{B}|$. The competing bid distribution of $B_t^{\mathsf{C}}$ is a discrete distribution over $\mathcal{B}$. Finally, the type of auction is also given as input. We allow for two types of auctions — first-price and second-price auctions. The distributions of $x_t(\cdot)$ and $q_t(\cdot)$ are fixed with these inputs of $B_t^{\mathsf{C}}$, $v_t$, and the auction type. We compare the performance of UCB-RoS against the approaches in [14, 18].

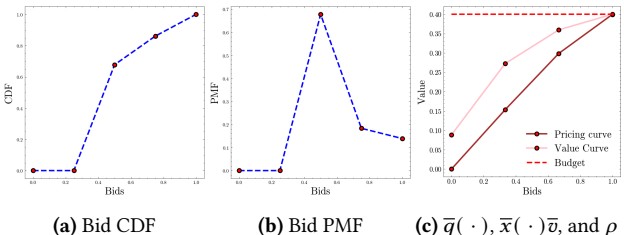

**(a)** Bid CDF      **(b)** Bid PMF      **(c)** $\overline{q}(\,\cdot\,), \overline{x}(\,\cdot\,)\overline{v}$, and $\rho$

**Figure 1:** Figures (a), (b) show the distribution over bids for the competing bidders. Figure (c) shows the expected pricing, expected value and budget curves over the bids.

The work of Castiglioni et al. [18] suggests a meta algorithmic game between a primal regret minimizing algorithm and a dual algorithm that minimizes the constraint violation. In our implementation of their algorithm, we choose the primal regret minimizer to be Exp3.P.1, as given in Auer et al. [6, Section 6], and the full information dual minimizer to be the DS-OMD algorithm, introduced in Fang et al. [25, Section 6]. The work of Bernasconi et al. [14] weights the constraint violation in a time decaying fashion and uses the primal regret minimizer EXP-IX of Neu [43].

We create a bidding instance with the parameters in Table 1 along with $w^*_{\text{LP}}$ and $V$ of the benchmark (2.3).

| Parameter | Value |
|---|---|
| $\mathcal{B}$ | $[0, 0.33, 0.66, 1]$ |
| $\rho$ | $0.4$ |
| $\overline{v}$ | $0.4$ |
| Auction type | Second Price |
| Value distribution | $\text{Beta}(10\overline{v}, 10(1-\overline{v}))$ |
| $w^*_{\text{LP}}$ | $[0, 0, 0, 1]$ |
| $V$ | $0.4$ |

**Table 1:** Table with parameter and benchmark values.

Figure 1(a), Figure 1(b) shows the distribution of the competing bids $(B^{\mathsf{C}}_t)$. This distribution has a mode at the bid $b = 0.333$. Figure 1(c) plots expected pricing $\overline{q}(\,\cdot\,)$ and realized value $\overline{x}(\,\cdot\,)\overline{v}$ as function of the bids. The bids with value and budget curves above the pricing curve are the feasible bids that satisfy the budget and RoS constraint in expectation. For the instance in Table 1, we see that all bids are feasible, and hence, the optimal $w^*_{\text{LP}}$ is $\delta_1(b)$. Thus, for this optimal allocation, the budget and RoS constraints are exactly satisfied. This suggests that both the constraints can be binding.

The experimental results are shown in Figure 3 for different horizons up to $2 \times 10^5$. Our algorithm, UCB-RoS (depicted in yellow), has a much smaller regret than those of Castiglioni et al. [18] (in green) and [14] (in blue). This primarily reflects our improved dependence on $|\mathcal{B}|$. The two baselines have much smaller constraint violations than UCB-RoS, which suggests that they each achieve a lower constraint violation at the cost of incurring near-linear regret. This near-linear regret for the chosen horizons is due to their worse dependence on $|\mathcal{B}|$. Hence, these baselines achieve sublinear regret

only over much larger horizons. In contrast, UCB-RoS achieves a much better trade-off between regret and constraint violation.

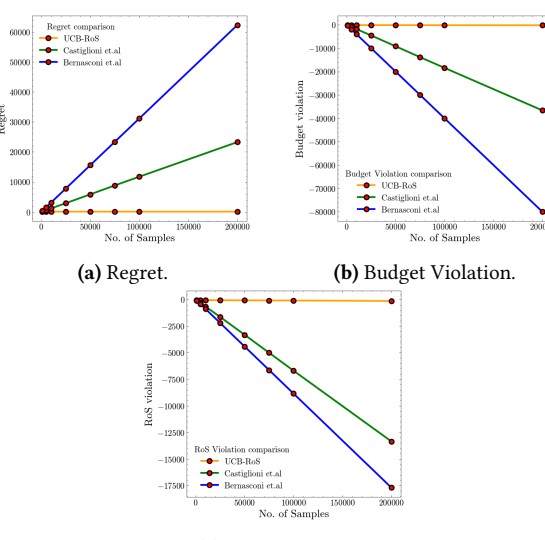

**(a)** Regret.      **(b)** Budget Violation.

**(c)** RoS Violation.

**Figure 2:** Comparison between UCB-RoS (in yellow), Castiglioni et al. [18] (in green), and Bernasconi et al. [14] (in blue).

We remark that the algorithms in the works of Castiglioni et al. [18] and Bernasconi et al. [14] were designed for *both* adversarial and stochastic rewards. Often, such algorithms are outperformed by algorithms designed for specific stochastic setting.

## 5 Conclusion and Future Work

In this paper, we studied online bidding with RoS and budget constraints when the value of an impression is unknown a priori. We developed a novel UCB-style algorithm that achieves near-optimal regret and constraint violation bounds without relying on restrictive assumptions like the existence of a Slater point. Our algorithm is not only theoretically sound but also computationally efficient. This work opens up several exciting avenues for future research. One direction is to extend our approach to more complex settings, such as those with multiple advertisers. Another promising direction is to consider adversarial environments where the competing bids or impression values are chosen adversarially. Finally, it would be valuable to develop variants of our algorithm that can incorporate contextual information into the decision-making process. We believe that our work takes a significant step towards developing more robust and effective bidding algorithms for online advertising.

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

# A Standard Concentration Results

We now state some well-known concentration inequalities that we use in our proofs.

*Fact A.1 (Hoeffding's inequality; Vershynin [48, Theorem 2.2.2]).* Let $X_1, X_2, \ldots X_N$ be independent symmetric Bernoulli random variables, and $a = (a_1, a_2, \ldots, a_N) \in \mathbb{R}^N$. Then, for any $t > 0$, we have

$$\mathbb{P}\left\{\sum_{i=1}^{N} a_i X_i \geq t\right\} \leq \exp\left(-\frac{t^2}{2\|a\|_2^2}\right).$$

*Fact A.2 (Line crossing inequality; Blackwell [15, Theorem 1]).* Let $X_0, X_1, X_2, \ldots$ be a martingale. Assume that: $|X_t - X_{t-1}| \leq 1$ almost surely for each $t$ and $X_0 = 0$, then for any $a, b > 0$ we have:

$$\mathbb{P}\{\exists t \in \mathbb{N} : X_t \geq a + bt\} \leq \exp(-ab).$$

*Fact A.3 (Azuma's inequality; Chung and Lu [20, Theorem 16]).* Let $X_0, X_1, X_2, \ldots$ be a martingale. Assume that: $|X_i - X_{i-1}| \leq c_i$ almost surely with $c_i > 0$ for each $i$. Then for any $t > 0$, we have

$$\mathbb{P}\{X_n - X_0 \geq t\} \leq \exp\left(-\frac{t^2}{2\sum_{i=1}^{n} c_i^2}\right).$$

## B  Bounding the Number of Wins

The following lemma relates $N_t$, the number of times the user won the bid until round $t$, with $w_t^*$. This is a crucial result we rely on to derive our regret bounds in Theorems 3.3, 3.4.

LEMMA B.1. *With probability at least $1 - \frac{2}{T}$, for every $t \in [T]$, it holds that*

$$\left|N_t - \sum_{s=1}^{t}\sum_{b \in \mathcal{B}} w_s^*(b)\overline{x}(b)\right| \leq \frac{2\log(T)}{V} + \frac{V \cdot t}{2}. \quad (B.1)$$

PROOF. We observe the fact that

$$\mathbb{E}_{t-1}[1_{\{N_t = N_{t-1}+1\}}] = \sum_{b \in \mathcal{B}} w_t^*(b)\overline{x}(b).$$

This implies that $N_t - \sum_{s=1}^{t}\sum_{b \in \mathcal{B}} w_s^*(b)\overline{x}(b)$ is a martingale with the increments bounded by 1. Applying the line cross inequality (Fact A.2) twice with $a = \frac{2\log(T)}{V}$ and $b = V/2$ gives the result. □

## C  High Probability Bounds

The goal of this section is to prove Theorem 3.4.

PROOF. From the proof of Theorem 3.3 we know that, with a probability at least $1 - \frac{3}{T}$, it holds that:

$$T \cdot V \leq \sum_{t=1}^{T} \mathbb{E}_{t-1}[v_t \cdot x_t(b_t)] + O\left(\sqrt{\frac{T\log(|\mathcal{B}|T)}{V}}\right). \quad (C.1)$$

Next, consider the martingale $X_t = \sum_{s=1}^{t} v_s \cdot x_s(b_s) - \mathbb{E}_{s-1}[v_s \cdot x_s(b_s)]$, with $X_0 = 0$. Clearly, we have $|X_t - X_{t-1}| \leq 1$. We can therefore invoke Fact A.3 on this martingale to get:

$$\mathbb{P}\left\{\sum_{t=1}^{T} v_t \cdot x_t(b_t) - \mathbb{E}_{t-1}[v_t \cdot x_t(b_t)] \geq \sqrt{2T\log(T)}\right\} \leq \frac{1}{T}.$$

Thus, we have that $\sum_{t=1}^{T} v_t \cdot x_t(b_t) - \mathbb{E}_{t-1}[v_t \cdot x_t(b_t)] \leq \sqrt{2T\log(T)}$ with a probability at least $1 - \frac{1}{T}$. Combining the earlier high probability bound with this via a union bound, we have that:

$$T \cdot V - O\left(\sqrt{\frac{T\log(|\mathcal{B}|T)}{V}}\right) \leq \sum_{t=1}^{T} v_t \cdot x_t(b_t)$$

with probability at least $1 - \frac{4}{T}$. This concludes the proof of the high probability regret bound. A similar analysis may be carried out for the constraint violation bounds of Theorem 3.3. To see this, first, applying Fact A.3 yields the following concentration statement:

$$\mathbb{P}\left\{\sum_{t=1}^{T} q_t(b_t) - \mathbb{E}_{t-1}[q_t(b_t)] \geq \sqrt{2T\log(T)}\right\} \leq \frac{1}{T}.$$

Combining this with the high probability bounds for budget and RoS violation under clean execution from the proof of Theorem 3.3, we have, with a probability at least $1 - 5/T$, the following:

$$\sum_{t=1}^{T} q_t(b_t) \leq \rho T + O(\sqrt{T\log(|\mathcal{B}|T)},$$

$$\sum_{t=1}^{T} q_t(b_t) \leq \sum_{t=1}^{T} v_t \cdot x_t(b_t) + O\left(\sqrt{\frac{T\log(|\mathcal{B}|T)}{V}}\right).$$

This concludes the proof of the high probability bounds on constraint violation for both budget and RoS constraints.

## D  Discussion on $V, \kappa$

In this section, we present a simple setting where $V = \Omega(1)$, but $\kappa = o(1)$. Let $B_t^{\max}$ be the highest bid of the competing bidders at round $t$. Let us denote the CDF of $B_t^{\max}$ as $F$ in this section. Let $b_0$ be a bid such that $0 < b_0 < 1$ and

$$F(b) > 0, \quad \forall b \geq b_0,$$

$$F(b) = 0 \quad \forall b < b_0.$$

Let us also assume the expected value $\overline{v} = b_0$, and the per-round budget $\rho \geq b_0$.

$\kappa = 0$. Consider the expected RoS constraint for a single round. For our choice of $\overline{v}$, it is easy to see that any bid $b \leq b_0$ satisfies the per-round RoS constraint in Equation (2.3), whereas any bid $b > b_0$ strictly violates the constraint (because $b > \overline{v}$). This is a setting where the Slater slack $\kappa = 0$.

$V = \Omega(1)$. Furthermore, in this setting it is easy to see that $V \geq x(b_0)b_0$ (by feasibility of $b_0$). We can choose $b_0$ so that $x(b_0)b_0 = \Omega(1)$. Thus, in this class, $V$ is bounded away from zero while the Slater slack $\kappa = 0$. This can happen in a practical scenario where the competing bidders have a distribution that often exceeds the value $\overline{v}$ with high probability.

In Figures 3 and 4 shows the worse performance of primal dual framework against UCB-RoS on this particular problem instance. This empirically verifies the theoretical claim above.

## E  Extension to Linear Bandits

The basic idea of maintaining optimistic sets for the constraints and playing an action from the resulting UCB problem can be extended to linear bandit setting. We discuss only briefly the pertinent aspects in this section.

### E.1  Setting

We consider the the linear bandit setting (see for for example chapter 19 in Lattimore and Szepesvári [35]). The basic elements of this setting are:

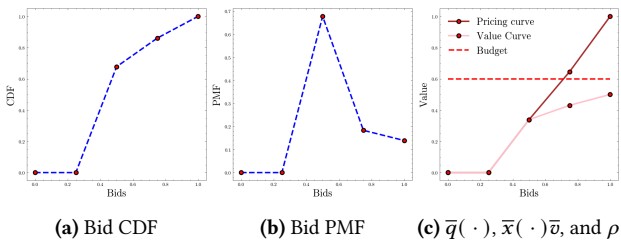

**(a)** Bid CDF     **(b)** Bid PMF     **(c)** $\overline{q}(\,\cdot\,)$, $\overline{x}(\,\cdot\,)\overline{v}$, and $\rho$

**Figure 3:** Figures (a), (b) show the distribution over bids for the competing bidders. Figure (c) shows the expected pricing, expected value and budget curves over the bids.

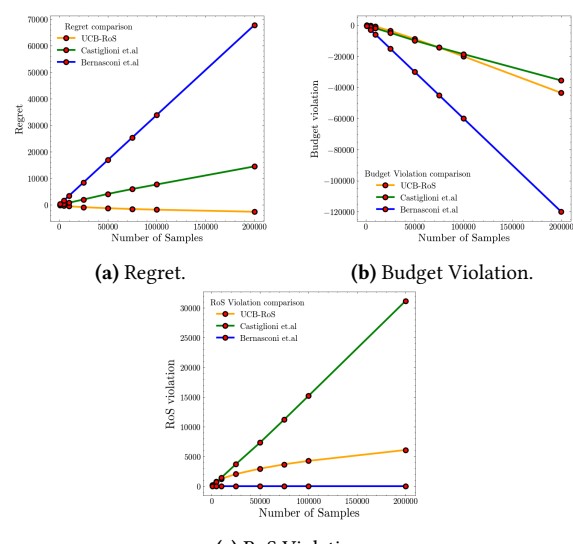

**(a)** Regret.     **(b)** Budget Violation.

**(c)** RoS Violation.

**Figure 4:** Comparison between UCB-RoS (in yellow), Castiglioni et al. [18] (in green), and Bernasconi et al. [14] (in blue).

(1) Loss observations and cost observations:

$$f_t(x_t) = \langle f, x_t \rangle + \epsilon_{t,x}, \quad g_{t,i}(x) = \langle g_i, x_t \rangle + \epsilon_{i,t,x},$$

where $\epsilon$ are 1-sub gaussian noise.

(2) $\|f\|_2 \leq B$ and $\|g_i\|_2 \leq B$. Here, $f, g_i$ are in $\mathbb{R}^d$.

(3) Action set $\mathcal{X}$ is a compact convex set of $\mathbb{R}^d$.

The agent pulls an arm $x_t$ at time $t$ and receives a noisy reward $f_t(x_t)$ and $i \in [m]$ constraint values $g_{t,i}(x_t)$. The goal is to minimize regret with respect to the stationary benchmark

$$x_{\text{OPT}} = \underset{x \in \mathcal{X}}{\text{argmax}} \langle f, x \rangle$$
$$\text{subject to } \langle g_i, x \rangle \leq 0, \quad \forall i \in [m],$$

while trying to ensure the constraint violation $-\sum_{t=1}^{T} g_{t,i}(x_t)$ is sublinear. The regret is formally defined as

$$\text{Regret} = T \cdot \langle f, x_{\text{OPT}} \rangle - \mathbb{E}\left[\sum_{t=1}^{T} f_t(x_t)\right].$$

## E.2 UCB-based Algorithm

We next describe various aspects of the UCB-style algorithm.

*OLS estimators*: The algorithm maintains a set of Ordinary Least Squares (OLS) estimators for $f, g_i$. At time $t$ we set the OLS estimators to be:

$$\widehat{f}_t := (\lambda I + \sum_{s=1}^{t} x_s x_s^T)^{-1} \sum_{s=1}^{t} f_s(x_s) x_s$$

$$\widehat{g}_t := (\lambda I + \sum_{s=1}^{t} x_s x_s^T)^{-1} \sum_{s=1}^{t} g_{s,i}(x_s) x_s.$$

These OLS estimators satisfy concentration inequalities. Let $V_t = \lambda I + \sum_{s=1}^{t} x_s x_s^T$, then we have that with probability $1 - \delta$:

$$\|\widehat{f}_t - f\|_{V_t} \leq \sqrt{d \log\left(1 + \frac{dB^2/\lambda}{\delta}\right)} + \lambda^{1/2} B$$

$$\|\widehat{g}_{i,t} - g_i\|_{V_t} \leq \sqrt{d \log\left(1 + \frac{dB^2/\lambda}{\delta}\right)} + \lambda^{1/2} B,$$

which can be derived using subgaussian concentration in a manner similar to Theorem 20.5 [35].

*Confidence sets*: Based on the above concentration inequalities one can derive confidence ellipsoids:

$$C_{f,t} := \left\{ f' \,\middle|\, \|\widehat{f}_t - f\|_{V_t} \leq \beta_t \right\}, \quad C_{g_i,t} := \left\{ g' \,\middle|\, \|\widehat{g}_{i,t} - g_i\|_{V_t} \leq \beta_t \right\},$$

such that, with high probability, $f, g_i$ lie in these sets. Here, we have $\beta_t = \sqrt{d \log\left(1 + \frac{dB^2/\lambda}{\delta}\right)} + \lambda^{1/2} B$.

*Algorithm*: For each time $t$, the algorithm repeats the following three steps sequentially:

(1) Action $x_t$ is chosen as follows:

$$x_{t+1} = \underset{x \in \mathcal{X}}{\arg\min} \quad \underset{f' \in C_{f,t}}{\min} \quad \underset{\substack{g_i'(x) \leq 0 \\ g_i' \in C_{g_i,t}}}{\min} \langle f', x \rangle. \quad (\text{E.1})$$

(2) Observe the noisy rewards $f_t(x_t)$ and the constraints $g_{i,t}(x_t)$. Update the OLS estimators and $V_t$ to incorporate the new data.

(3) Create confidence sets based on updated $V_t$ and OLS estimators.

*Remark* E.1. The setting of linear bandits with linear constraints that we consider here Equation (E.1) can, in general, be computationally very hard to solve without further structure in the problem.

## E.3 Regret and Constraint Violation Analysis

A regret and constraint violation bound may be derived by largely following the template of Theorem 3.3. The difference from the bidding problem is that, in this case, we do not have to derive concentration bounds for quantities like $N_t$.

*Regret analysis*: Assume that the minimizers in Equation (E.1) are $\overline{f}_t$ and $\overline{g}_{i_t}$, respectively. Further, we assume it holds with probability $1 - \delta_0$ (this can be done by choosing $\delta = \frac{\delta_0}{(m+1)T}$ and using union bound over $i, t$) that $f, g_i$ always belong in their respective confidence sets for all time $t \in [T]$. Then, by definition of the UCB choice, we have:

$$\langle f, x \rangle \geq \langle \overline{f}_t, x_t \rangle.$$

Hence, we have that:

$$r_t = \langle f, x_t \rangle - \langle f, x \rangle$$

$$\leq \langle f, x_t \rangle - \langle \overline{f}_t, x_t \rangle$$

$$\leq \|f - \widehat{f}_{t-1}\|_{V_{t-1}} \|x_t\|_{V_{t-1}^{-1}} + \|\overline{f}_t - \widehat{f}_{t-1}\|_{V_{t-1}} \|x_t\|_{V_{t-1}^{-1}}$$

$$\leq 2\beta_{t-1} \|x_t\|_{V_{t-1}^{-1}}.$$

Since $\beta_t$ is a increasing sequence and using the elliptical potential bound with log determinant (Theorem 19.3 in [35]), we have that:

$$\text{Regret} = \sum_{s=1}^{T} r_s \leq \sqrt{T \sum_{s=1}^{T} r_s^2}$$

$$\leq \sqrt{16T\beta_T^2 \log((\det(V_T)/\det(V_0)))}.$$

*Constraint violations*: We analyze the constraint violation next. Let $r_{s,i} = \langle g_i, x_s \rangle$. Then, we have that:

$$r_{t,i} \leq \langle g_i, x_s \rangle - \langle \overline{g_i}_t, x_t \rangle$$

$$\leq \|g_i - \widehat{g_i}_{t-1}\|_{V_{t-1}} \|x_t\|_{V_{t-1}^{-1}} + \|\overline{g_i}_t - \widehat{g_i}_{t-1}\|_{V_{t-1}} \|x_t\|_{V_{t-1}^{-1}}$$

$$\leq 2\beta_{t-1} \|x_t\|_{V_{t-1}^{-1}}.$$

Thus we get the following bound in the manner as before

$$\sum_{t=1}^{T} r_{t,i} \leq \sqrt{16T\beta_T^2 \log((\det(V_T)/\det(V_0)))}.$$

Further, $\log((\det(V_T)/\det(V_0))) \leq d \log \left(1 + TB^2/d\lambda\right).$ □

This gives a proof sketch for the following theorem.

**Theorem.** *The UCB-based algorithm in Appendix E.2, with $\lambda = \theta(1)$ and $\delta = 1/T$ in $\beta_t$, has a regret bound of*

$$\text{Regret} \leq \widetilde{O}(dB\sqrt{T})$$

*and constraint violation bounds of $\widetilde{O}(dB\sqrt{T})$ in expectation.*

Received 20 February 2007; revised 12 March 2009; accepted 5 June 2009

