# OpenReview forum: "Online bidding under RoS constraints without knowing the value"
_ACM.org/TheWebConf/2025/Conference — WWW 2025 Poster_

### Official Review · Reviewer_anHM · 2024-11-20

**Novelty:** 5
**Technical Quality:** 5

**Review:**

The authors consider the problem of bidding in online advertising when the value of each impression opportunity is unknown. The goal is to design an online bidding algorithm that maximizes the bidder’s revenue over the entire horizon, while respecting the RoS and budget constraints. The authors employ the UCB framework for estimating value, payment and allocation functions.The authors solve optimization problem (3.5), which determines the optimal probabilities for selecting each bid while accounting for budget constraints, RoS, and the fact that the auction parameters fall within their respective confidence intervals (solving this problem is also an important part of the work). The authors also empirically study the performance of UCB-RoS and compare it with other existing approaches on synthetically generated dataset.

This work is missing experimentation on real-world data, despite having a setup that appears conducive to such evaluations. Including real-data experiments would address several remaining issues and strengthen the paper.

**Questions:**

Please, correct 328-329 (paid price is : $x(b)\cdot p(b)$?)  (P.S. this remark had no impact on the assessment)

It appears that UCB estimates of $(x(), p(), v)$ rely on the assumption that the true parameter values are concentrated around the mean values. Is this correct? If so, it would be beneficial to explicitly mention this in the text. If not, will your approach yield robust results when value $v_t$ are NOT concentrated around a single value (0.4 for your synthetic problem in Experiment section)? For instance, consider a scenario (as part of your experiment) where values are drawn with equal probability from the discrete set $\{0.1, 0.4, 0.7\}$ (yet the mean value remains 0.4).

Will you evaluate the performance of UCB-RoS on publicly available dataset? (Experiments conducted using real-world data would significantly enhance this study, addressing many of the remaining issues).

**Reviewer Confidence:**

4: The reviewer is certain that the evaluation is correct and very familiar with the relevant literature

**Scope:**

4: The work is relevant to the Web and to the track, and is of broad interest to the community

---

### Official Review · Reviewer_pkJi · 2024-11-22

**Novelty:** 2
**Technical Quality:** 3

**Review:**

Quality: The paper is mathematically rigorous and provides clear theoretical bounds for regret and constraint violations. However, as the background of this paper aligns closely with actively researched state-of-the-art topics, it lacks significant innovation.
Clarity: The paper is somewhat difficult to read. Including a table for parameter notation would enhance clarity and improve the overall readability.
Originality: In my opinion, the background of this paper lacks originality, but how they address this common problem shows a degree of innovation.
Significance: The problem tackled in this paper is highly relevant to real-world scenarios, where uncertainty and constraints are critical factors. Therefore, I believe an evaluation based on real-world data is necessary to establish the paper's significance.
Pros:
1.	The paper tackles a highly relevant real-world problem in online auctions, where uncertainty and constraints play a critical role in this dynamic auction rounds.
2.	While the problem background lacks originality, the proposed method shows innovation in addressing a common issue.
3.	The use of confidence sets to handle unknown impression values while maintaining constraints is a notable contribution.
Cons:
1.	There are some inaccuracies in the paper, such as the expression “The experimental results are shown in Figure 3 for different horizons…,” which should instead be “The experimental results are shown in Figure 2…” if I am not mistaken.
2.	A table summarizing parameter notation and a figure illustrating the auction process are crucial for helping readers better understand the paper.
3.	Although the paper focuses on a more realistic scenario, the absence of experiments on real-world auction data significantly limits its practical relevance.
4.	The authors claimed that the proposed algorithm is applicable to both the first price and the second price auctions. However, they should have evaluated these two types of auctions respectively in their experiments to substantiate this claim.

**Questions:**

1. The regret and constraint violation bounds depend on V. Can the authors provide more insight into how V is expected to behave in realistic advertising scenarios?
2. Have the authors considered the computational cost in environments with hundreds of competitors?
3. Have the authors tested their algorithm on real-world advertising datasets? What is the expected performance in these settings?
4. The paper focuses on RoS and budget constraints. Could the proposed framework be generalized to handle additional constraints, such as maximum frequency caps?

**Reviewer Confidence:**

3: The reviewer is confident but not certain that the evaluation is correct

**Scope:**

3: The work is somewhat relevant to the Web and to the track, and is of narrow interest to a sub-community

---

### Official Review · Reviewer_k6Qq · 2024-11-30

**Novelty:** 4
**Technical Quality:** 5

**Review:**

Summary:

This paper studies a common problem of learning to bid in online advertising auctions with RoS and budget constraint. The main difference in this work is that they assume that the bidder does not know their value for an impression unless they win. This leads to an exploration-exploitation trade-off that requires a UCB style algorithm to solve.

They end with an experimental section that shows in synthetic experiments the UCB based algorithm they develop achieves better regret and lower violations to constraints than algorithms from other settings.

Overview:

In general, I think the paper is well written and easy enough to follow. The setting is compelling but I have a few hesitations about the model I will list below.

Strengths

- The online bidding setting where advertiser do not know their value is a compelling and natural setting. Advertiser often experiment with high bids in the market and this gives a reasonable explanation for this phenomenon.

- UCB seems like a natural approach and they achieve good regret bounds.

- The paper is well written and provides good intuition.

Weaknesses

- While the paper has an extensive discussion about related work I found it hard to parse exactly what the state of the art result was for this setting prior to the authors work.

- This setting captures an IPV model of bidders values. In the real world advertisers usually have some sort of demographic and cookie information before placing their bids. I would have liked to see a bit of discussion of whether the results carry over at all in the case where bids are conditionally independent.

- If I am reading this correctly, in general the bidders observe what their allocative probability was even in the case where they do not win the impression. I don't understand why this assumption makes sense. I suppose it doesnt matter in the case of FPA and SPA but for the general results on monotone allocation rules it seems strange. Its possible I am just misunderstanding and happy to remove this if so.

**Questions:**

Questions:

1. Does the advertiser observe the allocation probability even in case when they do not win? If so why does this assumption make sense?

2. Do you have any intuition about whether the results would substantially change in a model where advertisers are conditionally independent given some information set assuming the have global budget and RoI constraints?

3. It is typical for advertising platforms to share the highest other bid with bidders after the auction. Would this substantially change the results?

4. Is there a way to convert this algorithm to one that satisfies budget and RoI constraints with high probability at the cost of some additional regret?

**Reviewer Confidence:**

3: The reviewer is confident but not certain that the evaluation is correct

**Scope:**

4: The work is relevant to the Web and to the track, and is of broad interest to the community

---

### Official Review · Reviewer_ryeb · 2024-12-02

**Novelty:** 5
**Technical Quality:** 4

**Review:**

The paper addresses online advertising bidding with unknown impression values by proposing a novel UCB-style algorithm that balances exploration and exploitation while maintaining budget and RoS constraints. Prove their algorithm for both regret and constraint violation - the first near-optimal bounds in this setting - and demonstrate its effectiveness through experimental validation.

Pros: 1.Novel approach to online bidding with RoS constraints without knowing impression values, addressing a practical problem. 2.Removes restrictive assumptions like Slater conditions present in existing works, enhancing algorithm applicability.

Cons: 1. Multiple bidder scenarios not addressed 2. Lack of adversarial environment analysis

**Questions:**

1.How does the system converge when multiple advertisers simultaneously use the UCB-RoS algorithm?  2.Is it possible to design versions that adapt to dynamic constraints?  3. How do interactions between multiple constraints affect algorithm performance?

**Reviewer Confidence:**

1: The reviewer's evaluation is an educated guess

**Scope:**

3: The work is somewhat relevant to the Web and to the track, and is of narrow interest to a sub-community

---

### Official Review · Reviewer_PQFo · 2024-12-03

**Novelty:** 5
**Technical Quality:** 5

**Review:**

The paper tackles the challenge of online advertising bidding where an advertiser must maximize value under budget and Return-on-Spend (RoS) constraints without prior knowledge of the value generated by each impression. The authors propose a novel Upper Confidence Bound (UCB)-style algorithm that balances exploration and exploitation to learn the optimal bidding strategy and impression values concurrently. The algorithm's performance is evaluated through theoretical analysis, demonstrating an optimal regret and constraint violation bound, and empirical validation on synthetic data.

Advantages:
(1) The problem addressed is highly relevant to the online advertising industry, and the solution provided is both theoretically sound and practically applicable, making it valuable for both academia and industry practitioners;
(2) The paper is well-structured, with clear definitions, a logical flow of ideas, and a comprehensive discussion of related work. The use of figures and tables to illustrate the algorithm's performance is effective;
(3) The introduction of related work effectively highlights the originality of this paper.

Disadvantages:
(1) While the algorithm is presented as computationally efficient, the paper could benefit from a deeper discussion on the practical challenges of implementing the algorithm in real-world online advertising platforms;
(2) The theoretical proofs in the main text are overly complex. It might be helpful to provide a more concise and understandable outline in the main body.

**Questions:**

Here are some advice for the limitations of the paper:
(1) Including case studies or experiments with real-world data would strengthen the paper's claims and demonstrate the algorithm's effectiveness in actual scenarios. Simulated experiments seems too weak for this paper;
(2) Comparing the UCB-style algorithm with other online learning frameworks could highlight its unique strengths and potential limitations;
(3) A sensitivity analysis on the choice of parameters and their impact on the algorithm's performance would provide insights into its robustness and practical usability.

**Reviewer Confidence:**

4: The reviewer is certain that the evaluation is correct and very familiar with the relevant literature

**Scope:**

4: The work is relevant to the Web and to the track, and is of broad interest to the community